# Reproducibility of Portable OCT and Comparison with Conventional OCT

**DOI:** 10.3390/diagnostics14131320

**Published:** 2024-06-21

**Authors:** Marie Nakamura, Takao Hirano, Yoshiaki Chiku, Yoshiaki Takahashi, Hideki Miyasaka, Shinji Kakihara, Ken Hoshiyama, Toshinori Murata

**Affiliations:** Department of Ophthalmology, Shinshu University School of Medicine, Matsumoto 390-8621, Japan

**Keywords:** portable, optical coherence tomography (OCT), mean retinal thickness

## Abstract

Optical coherence tomography (OCT) is an indispensable instrument in ophthalmology; however, some facilities lack permanent OCT devices. ACT100, a portable SD-OCT system, allows for medical examinations at hospitals that do not have OCT and house calls. We investigated the usefulness of ACT100 by examining the reproducibility of retinal thickness measurements in 35 healthy participants with normal eyes using ACT100 and Cirrus. Using two OCTs, the OCT imaging of both eyes of each subject was performed. Macular retinal thickness was evaluated using the average value in nine lesions of the Early Treatment Diabetic Retinopathy Study (ETDRS) circle. Both models captured images in all cases. In the right eye, mean retinal thickness was significantly lower than in the ACT100 group in all regions; however, the measured values correlated well. The intraclass correlation coefficients showed the same high reliability as the Cirrus. The coefficients of variation (CVs) of both models showed little variation and high stability; however, the CV of ACT100 was significantly higher. The left eye was almost identical. Macular retinal thickness measured using ACT100 showed slightly greater variability than that by Cirrus; the reproducibility was good and correlated well with that of Cirrus. This technique is a suitable alternative to conventional OCT.

## 1. Introduction

Optical coherence tomography (OCT) can noninvasively produce tomographic images of ocular tissues. The clinical use of OCT in ophthalmology began in 1997 when Humphrey (Carl Zeiss, Baden-Württemberg, Germany) released OCT2000, a time-domain OCT. Since then, various companies have released various OCT devices, and their resolution and imaging speed have significantly improved owing to a shift from the time domain to the spectral domain method. The recently introduced swept-source OCT uses a tunable wavelength light source. This technology has a higher penetration ability in deeper layers as well as in more opaque optical media [1]. As a result, OCT is now becoming an indispensable instrument in the daily practice of ophthalmology [2,3,4,5,6,7,8,9]. However, some facilities do not have permanently installed OCT devices because of their high cost and stationary nature. Additionally, taking images with conventional OCT requires patients to remain seated for a certain period of time, which makes it difficult to perform OCT examinations on patients who have difficulty maintaining their posture or transferring from one position to another. In recent years, portable OCT devices have emerged as a solution to this problem, and their usefulness in screening for age-related macular degeneration and in pediatric care has been reported [10,11]. However, most of these reports are based on qualitative evaluations, and reports on quantitative evaluations are lacking. The ACT100 ver.19 (MiiS, Hsinchu, Taiwan), a lightweight and compact portable SD-OCT system consisting of a main box (234 × 190 × 108 mm, 3.35 kg) and a probe (275 × 90 × 74 mm, 0.75 kg), was launched in 2023. This OCT system is expected to be useful in clinical settings, such as medical examinations at facilities without conventional OCT equipment or during house calls, because it enables both qualitative and quantitative evaluation. However, the accuracy of this system in actual clinical situations has not been evaluated, and a comparison with existing OCT systems has not yet been conducted. In this study, we investigated the usefulness of portable OCT by examining the reproducibility of retinal thickness measurements in normal eyes using portable OCT and an existing stationary OCT (Cirrus 5000 HD-OCT plus, Carl Zeiss, Baden-Württemberg, Germany) and compared the results between the two models.

## 2. Subjects and Methods

### 2.1. Study Population

This cross-sectional, observational study was reviewed and approved by the Ethics Committee of Shinshu University (approval No. 5923). All studies conducted adhered to the tenets of the Declaration of Helsinki. Written informed consent was obtained from all the subjects. All subjects were recruited at the Department of Ophthalmology at Shinshu University Hospital in May 2023. Subjects were only included if they were healthy and had normal eyes, i.e., if they had a best corrected visual acuity of 20/20 or better, and without any of the exclusion criteria. Subjects were excluded if any of the following were present: significant refractive error (myopia more severe than −6.00 diopters or hyperopia more severe than +3.00 diopters), ocular media opacity, or retinal disease.

### 2.2. Image Acquisition and Evaluation

Using two OCT systems, a portable OCT system (ACT100, MiiS, Hsinchu, Taiwan) and an existing stationary OCT system (Cirrus 5000 HD-OCT plus, Carl Zeiss, Baden-Württemberg, Germany), OCT imaging of both eyes of each subject was performed on the same day without mydriasis by one examiner familiar with the testing method. ACT100 is a lightweight and compact portable OCT system consisting of a main box (234 × 190 × 108 mm, 3.35 kg) and a probe (275 × 90 × 74 mm, 0.75 kg) (Figure 1a). Accessories such as cables and monitors can be stored in a single carrying case. The carrying case was 290 mm × 280 mm × 520 mm, 13.5 kg, and could be rolled and moved like a general carrying case (Figure 1b). It is a spectral-domain OCT with a scan speed of 80,000 A-scan/s, a light source wavelength of 840 nm, a lateral resolution of 20µm, and a longitudinal resolution of 10 µm. On the other hand, the Cirrus HD-OCT plus is a spectral-domain OCT with a scan speed of 27,000–68,000 A-scan/s, a light source wavelength of 840 nm, a lateral resolution of 15 µm, and a longitudinal resolution of 5 µm. A comparison of the features of the two devices is presented in Table 1. Macular retinal thickness was evaluated using the average value of the Early Treatment Diabetic Retinopathy Study (ETDRS) circle, which was automatically evaluated using the analysis software of each device. We analyzed nine subfields within the ETDRS circle. The foveal central subfield was defined as the inner 1-mm-diameter circle; the pericentral subfield was located between the inner and middle 3-mm-diameter circles; and the peripheral subfield was between the middle and outer 6-mm-diameter circles. Based on a previous report, the pericentral and peripheral areas were further divided into four quadrants: superior, temporal, nasal, and inferior [12] (Figure 2). The measurement protocols used for each device were as follows:

Cirrus: 6.0 × 6.0 mm area captured with macular cube scan. Each image consisted of 512 A scans × 128 scans in the horizontal direction.

ACT100: Thickness map-retina scan of 9.0 × 9.0 mm. Each image contains 512 A scans × 129 horizontal scans. Images were taken until three images with a confidence coefficient automatically measured by each instrument (Cirrus: signal strength ≥6, ACT100: signal index ≥6) were obtained, and the number of images taken was compared between the two models.

### 2.3. Statistical Analyses

Statistical analyses were performed using the Statistical Package for Social Sciences version 22.0 (IBM, Armonk, NY, USA). Continuous variables were expressed as mean values ± standard deviation. The reproducibility was examined using the intraclass correlation coefficient (ICC1,1). The coefficient of variation (CV) obtained for each instrument was compared. Pearson’s correlation coefficient was used to determine the correlation between the two models. CVs were multiplied by a factor of 1000. Statistical significance was set at *p* < 0.05. Each case was analyzed separately for the right and left eyes to exclude case-specific bias.

## 3. Results

Thirty-five participants (18 males and 17 females, 37.9 ± 10.8 years) were included in the final analysis. Both models were capable of capturing images in all the cases. In the right eye, the number of images taken (times) before three images with a sufficient confidence coefficient were obtained was significantly higher for ACT100 (3.2, 3.9, *p* = 0.0057). Mean retinal thickness was significantly lower in the ACT100 group in all regions; however, the measured values were correlated (Table 2). The mean retinal thickness of the fovea was about 10 µm lower in ACT100 than in Cirrus, and measurement differences were observed in other regions, as shown in Table 2. The ICC was lower than 0.9 in the right peripheral superior, peripheral inferior, and peripheral nasal regions of the ACT100 but higher than 0.9 in the other regions, showing the same high reliability as that of the Cirrus (Table 3). The CVs of both models showed little variation and high stability; however, the CV of the ACT100 was significantly higher than that of the Cirrus (Table 3).

In contrast, in the left eye, there was no significant difference between the two models in the number of images captured before the three images with sufficient confidence coefficients were obtained (3.2 vs. 3.5, *p* = 0.1489). The mean retinal thickness was significantly lower in the ACT100 group in all regions; however, there was a correlation between the measured values (Table 4). The mean retinal thickness of the fovea was about 10 µm lower in ACT100 than in Cirrus, as in the right eye, and the measurement differences in other areas are as shown in Table 4. The ICC of the ACT100 was 0.9 or higher in all regions, offering the same high reliability as that of the Cirrus (Table 5). The CVs of both models showed slight variation and high stability at all sites; however, the CV of ACT100 was significantly higher than that of Cirrus (Table 5).

## 4. Discussion

The present study evaluated the measurement reproducibility of ACT100, a portable OCT with an unprecedented portability concept, in normal eyes and compared its accuracy with that of Cirrus, a conventional OCT. In all cases, sufficient OCT images in clinical settings were obtained using ACT100. Although slightly inferior to Cirrus in terms of reproducibility, the three measurements showed a high degree of agreement. In all nine ETDRS subfields, although the retinal thickness measured using the ACT100 was lower than that measured using Cirrus, the measurements using the two models were highly correlated. Despite being portable, this study revealed that the ACT100 has high reproducibility and accuracy in measuring retinal thickness in normal eyes, although it is somewhat inferior to conventional OCT. 

Three satisfactory images were obtained for both eyes in all the cases. However, the number of images captured until three images with sufficient confidence coefficients were obtained was significantly higher with ACT100 in the right eye. On the other hand, no significant difference was observed for the left eye. The ACT100 is a newer device; therefore, some time was required to get used to it. In this study, all images were taken from the right eye, and it is considered that the left eye could be photographed less frequently than the right eye, which is the same as the Cirrus, owing to the habituation of the examinees and test subjects. In this study, all the images were acquired from the right eye. Therefore, habituation to the examination may decrease the number of examinations required to capture left-eye images. In addition, when examining the right eye, the probe was difficult to examine in some cases because it rested against the nose. This may be one of the reasons for this finding.

Next, regarding the reliability of the measurement, the ICC of the ACT100 was 0.9 or higher in almost all regions, showing high reliability equivalent to that of the Cirrus. However, the peripheral superior, peripheral inferior, and peripheral nasal regions of the right eye were lower than those of the other regions. The CVs of both models exhibited little variation and high stability; however, the CV of ACT100 was significantly higher than that of Cirrus. One of the reasons why the CV of ACT100 is higher than that of Cirrus is the longer acquisition time. According to the manufacturer’s announcement, the time required for the ACT100 was approximately 2.5 s, whereas that for the Cirrus was approximately 1.5 s. This was expected to increase the subject’s fatigue level and cause unstable fixation, resulting in a larger CV than that of Cirrus. However, the results of this study indicate that, even at this time, the reproducibility of ACT100 was satisfactory and useful enough for clinical evaluation, although the ICC was lower and the CV was slightly higher than those of Cirrus.

In many phase III trials for age-related macular degeneration, retinal vein occlusion, and diabetic macular edema [13,14,15,16,17], central macular thickness (CMT) values have been used as the criteria for the administration of anti-VEGF therapy to provide quantifiable criteria for treatment. CMT values are also important in daily practice as they serve as a reference for treatment. Evaluating the macular retinal thickness in the same eye using the two instruments showed that the measured values correlated. However, the ACT100 showed significantly lower values in all areas of both eyes. The CMT of both eyes was about 10 µm lower in ACT100 than in Cirrus, and measurement differences were observed in all other subfields. Several previous reports have discussed the CMT measurement error of each instrument [18,19,20,21,22,23,24,25,26,27]. It has been reported that the measurement difference in CMT is related to the difference in the retinal measurement range of each device [18,19,20,21]. In a study comparing Cirrus and Spectralis (Heidelberg Engineering, Heidelberg, Germany), Cirrus measured the center of the retinal pigment epithelium (RPE) from the inner limiting membrane (ILM) when evaluating retinal thickness. Spectralis measures the lower edge of the RPE from the ILM, which results in a thicker CMT [18]. However, the ACT100 measures from the ILM to the center of the RPE, similar to the Cirrus, and no difference in the measurement range was observed between the two models. Other possible causes of differences in retinal thickness measurement include differences in the scan interval, refractive index, and longitudinal resolution. Suzuma et al. reported that the difference between the A- and B-scan intervals might affect the difference in CMT measurements [21]. Cirrus used 200 scans with 200 A-scans per scan within an area of 6.0 × 6.0 mm, and Spectralis used 19 scans with 512 A-scans per scan within an area of 6.0 mm × 4.5 mm, which shows that the scan spacing is different among the models. The possibility that the Spectralis model, which has more spaced B-scans, may not be able to capture detailed retinal lesions, resulting in differences in CMT measurements, was discussed. The A-scan of both Cirrus and ACT100 used in this study is 512, and the B-scan is almost the same, 128 and 129, but the measurement range is different, 6.0 × 6.0 mm vs. 9.0 × 9.0 mm. Because ACT100 measures a wider range, the interval between scans was broader than that of Cirrus. Therefore, this may have affected the measurement differences. The refractive index required to convert the optical delay measured using OCT to an equivalent value is 1.379 for ACT100, which has not been reported for Cirrus. The refractive indices vary for different models, which may have affected the measurement difference. In addition, the significant difference in longitudinal resolution between ACT100 (10 µm) and Cirrus (5 µm) might also affect the measurement difference. Although the reasons mentioned above may account for the difference in retinal measurements between the ACT100 and Cirrus, users need to know the characteristics of each OCT system. 

In our recent memory, there have been cases where patients were reluctant to visit hospitals due to the coronavirus pandemic, which prevented them from receiving appropriate treatment or caused them to miss the appropriate time for treatment [28,29,30]. If such a mobile OCT device is available, it will be possible to evaluate the presence or absence of disease and disease exacerbation, even in local hospitals or during house calls, which may lead to appropriate treatment. Although portable OCT can capture images with the same high reproducibility as existing OCT, some modifications are necessary when capturing images of patients with difficulty in fixation, narrow pupil size, elderly patients with cataracts, or other conditions that cause strong intermediate translucent zone opacities. In this study, we captured and analyzed images of healthy subjects, and we are now evaluating whether we can use this technique in diseased eyes. 

This study had some limitations. First, it was conducted only in normal eyes without retinal disease; therefore, future studies focusing on diseased eyes are needed. Measurement reproducibility was slightly inferior for ACT100 compared with Cirrus. We believe that the handheld nature of ACT100 might cause some rotation, reducing reproducibility. According to the marketing company, this probe is being improved to reduce possible rotations, and higher measurement reproducibility is expected. ACT100 Ver. 20 is currently under development, and improvements in the software may reduce the measurement error. In this study, only retinal thickness was evaluated. According to recent reports, distinct fluid volume measured using artificial intelligence (AI)-based image analysis is a much more accurate and functionally related marker than retinal thickness [31]. It is expected that the usefulness of ACT100 used in this study will be enhanced if distinct fluid volume can be measured using AI-based image analysis, and this is an issue for future studies.

Although the CMT measured using ACT100 showed slightly greater variability than that measured using Cirrus, the reproducibility was good and correlated well with that measured using Cirrus. This technique may be an alternative to conventional OCT in facilities that do not require OCT or in-house calls.

## Figures and Tables

**Figure 1 diagnostics-14-01320-f001:**
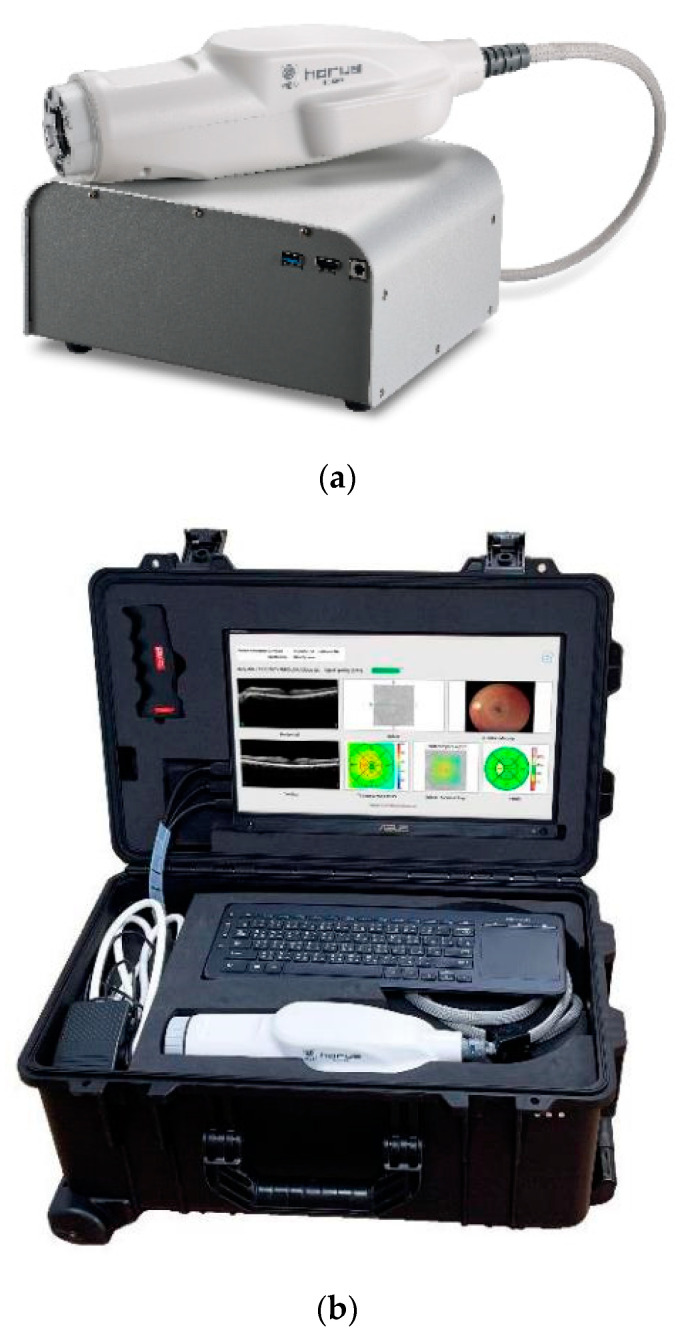
(**a**) A main box and a probe of ACT100. (**b**) The carrying case of ACT100.

**Figure 2 diagnostics-14-01320-f002:**
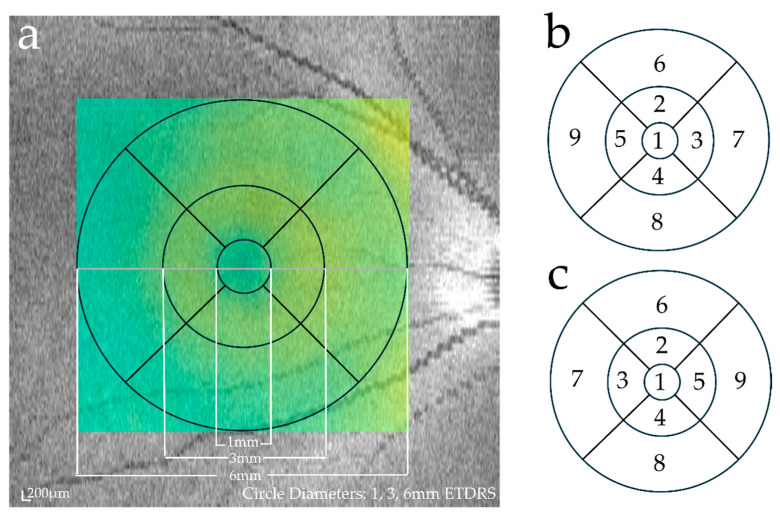
Early Treatment Diabetic Retinopathy (ETDRS) grid. (**a**) Delineation of the nine macular sectors, according to the ETDRS, within which we measured macular layer thickness. (**b**) Nine ETDRS sectors in the right eye. (**c**) Nine ETDRS sectors in the left eye. Green indicates that the retina is of normal thickness.

**Table 1 diagnostics-14-01320-t001:** Comparison of the features of the two devices.

	Cirrus	ACT100
Manufacturer	Carl Zeiss	MiiS
Principles of OCT	Spectral-domein OCT	Spectral-domein OCT
Scan speed (A-scan/s)	27,000–68,000	80,000
Light source wavelength (nm)	840	840
Lateral resolution (µm)	15	20
Longitudinal resolution (µm)	5	10

**Table 2 diagnostics-14-01320-t002:** Mean retinal thickness using ACT 100 and Cirrus OCT in right eyes and the differences with Pearson’s correlation coefficients for each ETDRS subfield.

	Mean Retinal Thickness (μm)				Pearson’s Correlation Coefficient	
	Cirrus	ACT100	Difference	*p*	Pearson’s Correlation Coefficient	*p*
Fovea	257.9	245.8	12.1	<0.0001	0.9295	<0.0001
Pericentral superior	330.1	289.5	40.6	<0.0001	0.8271	<0.0001
Pericentral temporal	316.8	284.2	32.5	<0.0001	0.6086	0.0001
Pericentral nasal	322.9	282.6	40.4	<0.0001	0.8350	<0.0001
Pericentral inferior	330.5	296.5	34.0	<0.0001	0.8175	<0.0001
Peripheral superior	286.0	259.5	26.6	<0.0001	0.3794	0.0246
Peripheral temporal	266.2	244.5	21.7	<0.0001	0.7903	<0.0001
Peripheral inferior	268.1	255.1	13.0	<0.0001	0.6985	<0.0001
Peripheral nasal	304.1	281.9	22.2	<0.0001	0.6030	0.0001

ACT 100 and Cirrus OCT results were compared using paired-*t* test.

**Table 3 diagnostics-14-01320-t003:** ICC of three consecutive measurements of retinal thickness using ACT 100 and Cirrus OCT in right eyes and the CV for each ETDRS subfield.

	ICC		CV		
	Cirrus	ACT100	Cirrus	ACT100	*p*
Fovea	0.998	0.986	5.4	13.6	<0.0001
Pericentral superior	0.99	0.947	5.6	13	<0.0001
Pericentral temporal	0.994	0.965	4.2	12.4	<0.0001
Pericentral nasal	0.995	0.926	4.1	18.9	<0.0001
Pericentral inferior	0.992	0.933	5.4	18.1	<0.0001
Peripheral superior	0.95	0.686	9.4	23.6	0.0036
Peripheral temporal	0.993	0.921	5.6	19	<0.0001
Peripheral inferior	0.985	0.717	6.5	26.1	<0.0001
Peripheral nasal	0.994	0.795	4.5	24.9	<0.0001

ICC, intraclass correlation coefficient. CV, coefficient of variation. ACT 100 and Cirrus OCT results were compared using Wilcoxon signed-rank test.

**Table 4 diagnostics-14-01320-t004:** Mean retinal thickness using ACT 100 and Cirrus OCT in left eyes and the differences with Pearson’s correlation coefficients for each ETDRS subfield.

	Mean Retinal Thickness (µm)				Pearson’s Correlation Coefficient	
	Cirrus	ACT100	Difference	*p*	Pearson’s Correlation Coefficient	*p*
Fovea	258.6	247.5	11.1	<0.0001	0.8970	<0.0001
Pericentral superior	329.5	291.5	38.0	<0.0001	0.8644	<0.0001
Pericentral temporal	316.0	285.9	30.1	<0.0001	0.8179	<0.0001
Pericentral nasal	323.4	283.8	39.6	<0.0001	0.8469	<0.0001
Pericentral inferior	331.9	299.6	32.3	<0.0001	0.8885	<0.0001
Peripheral superior	284.5	262.0	22.5	<0.0001	0.8042	<0.0001
Peripheral temporal	264.3	247.0	17.2	<0.0001	0.7142	<0.0001
Peripheral inferior	267.3	251.0	16.3	<0.0001	0.7640	<0.0001
Peripheral nasal	303.7	280.9	22.8	<0.0001	0.8703	<0.0001

ACT 100 and Cirrus OCT results were compared using paired-*t* test.

**Table 5 diagnostics-14-01320-t005:** ICC of three consecutive measurements of retinal thickness using ACT 100 and Cirrus OCT in left eyes and the CV for each ETDRS subfield.

	ICC		CV		
	Cirrus	ACT100	Cirrus	ACT100	*p*
Fovea	0.998	0.983	4.4	13.8	<0.0001
Pericentral superior	0.997	0.961	3.2	13.8	<0.0001
Pericentral temporal	0.996	0.972	3.5	11.3	<0.0001
Pericentral nasal	0.995	0.955	3.9	15.2	<0.0001
Pericentral inferior	0.997	0.959	3.0	13.6	<0.0001
Peripheral superior	0.988	0.965	5.5	12.3	0.0036
Peripheral temporal	0.984	0.906	5.1	19.4	<0.0001
Peripheral inferior	0.996	0.907	4.8	18.2	<0.0001
Peripheral nasal	0.998	0.964	2.9	10.3	<0.0001

ICC, intraclass correlation coefficients. CV, coefficient of variation. Comparisons between ACT 100 and Cirrus OCT were carried out with Wilcoxon signed-rank test.

## Data Availability

The data supporting the findings of this study are available from the corresponding author, M.N., upon reasonable request.

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
