# Peer review of "Reproducibility of Portable OCT and Comparison with Conventional OCT"

_diagnostics, 2024, doi:10.3390/diagnostics14131320_

Round 1

Reviewer 1 Report

Comments and Suggestions for Authors

Some weakness points and inappropriate as follows:

"1. Introduction" is too short. Please add references regarding the portable OCT. "p.2, 2.2 Image Acquisition and Evaluation", line 81: what is the definition of confidence coefficients? How do you calculate?

p.3: the caption of Figure 1 should be consistent with the figure content. Is it "a b" or "A B"?

p.6: line136, is Table 5 ? or should it be Table 6?

Table 6: Is the fovea thickness using ACT100 only 27.5? Compared to 258.6 in Cirrus, the difference cannot be 11.1, as Table 6 shows.

4. Discussion (p.8), lines 156-157: How can the accuracy of measuring the retinal thickness of the ACT100 be defined? Since ACT100 has two times worse longitudinal resolution than Cirrus, how can authors compare the accuracy in thickness measurement, and what will the difference range be accepted in clinics? 

Lines 178-179: If the acquisition time for ACT100 is 9.2 seconds, and the total A-scans number (129*512, mentioned on page 2), the scan speed will be around 7k (A-scan/sec), which is not consistent with 80k shown in Table1. 

4. Discussion (p.9), lines 201-203: the possible causes of differences in retinal thickness measurement should include differences in the "longitudinal resolution". Lines 217-219: The OMAG algorithm used by Cirrus is only related to the calculation of angiography, which will not affect the thickness measurement. Lines 235-236: Why measurement reproducibility can be influenced by the shape of the probe?

The above content descriptions were unclear or confusing; please rewrite them to make them understandable. 

Comments on the Quality of English Language

Abstract: line 17, what are "CVs"? Writing the full name before introducing an abbreviation is essential.

2.2 Image Acquisition and Evaluation: line77, "6.0X6.0 mm area" should be mm2

4. Discussion (p.8), lines 159-162: The sentence is too long and hard to understand; line 176, please correct "ofCirrus". P.9. lines 213-215. The descriptions of the content were unclear. Is "1.379" the thickness or refractive index?

4. Discussion (p.9), line 227: Although "OCT"....., here could use "portable OCT" for comparing to existing OCT.

Author Response

May 30th, 2024

Editor and Reviewers

Diagnostics

Thank you very much for reviewing our manuscript and providing excellent comments to help us improve the paper. We have revised the manuscript in accordance with your recommendations. Please find below point-by-point responses to the reviewers’ comments, along with corresponding changes made in the manuscript. The changes are indicated in red in the revised manuscript for your convenience.

Reviewer #1

Comment 1. "1. Introduction" is too short. Please add references regarding the portable OCT. "p.2, 2.2 Image Acquisition and Evaluation", line 81: what is the definition of confidence coefficients? How do you calculate?

Authors’ reply: Thank you for the valuable advice. As you pointed out, we believe that it would be useful for readers to share previously reported information on portable OCT. Confidence coefficients are automatically calculated for each device. We have included detailed explanations and relevant references in the revised Introduction section as follows:

(Original)

Optical coherence tomography (OCT) can noninvasively produce tomographic images of ocular tissues. The clinical use of OCT in ophthalmology began in 1997 when Humphrey (Carl Zeiss, Baden-Württemberg, Germany) released OCT2000, a time-domain OCT. Since then, various companies have released various OCT devices, and their resolution and imaging speed have significantly improved owing to a shift from the time domain to the spectral domain method. OCT is now becoming an indispensable instrument in the daily practice of ophthalmology [1–8]. However, some facilities do not have permanently installed OCT devices because of their high cost and stationary nature. In addition, conventional OCT requires the patient to remain sitting for a certain time during imaging, making it difficult to maintain their posture or transfer to another position for imaging. The ACT100 ver.19 (Miis, Hsinchu, Taiwan), a lightweight and compact portable SD-OCT system consisting of a main box (234 × 190 × 108 mm, 3.35 kg) and a probe (275 × 90 × 74 mm, 0.75 kg), was launched in 2023. This OCT system is expected to be useful in clinical settings, such as medical examinations at facilities without conventional OCT equipment or during house calls. However, the accuracy of this system in actual clinical situations has not been evaluated, and a comparison with existing OCT systems has not yet been conducted. In this study, we investigated the usefulness of portable OCT by examining the reproducibility of retinal thickness measurements in normal eyes using portable OCT and an existing stationary OCT (Cirrus HD-OCT plus, Carl Zeiss) and compared the results between the two models.

Images were taken until three images with a confidence coefficient (Cirrus: signal strength ≥ 6, ACT100: signal index ≥ 6) were obtained, and the number of images taken was compared between the two models.

(Revised)

Optical coherence tomography (OCT) can noninvasively produce tomographic images of ocular tissues. The clinical use of OCT in ophthalmology began in 1997 when Humphrey (Carl Zeiss, Baden-Württemberg, Germany) released OCT2000, a time-domain OCT. Since then, various companies have released various OCT devices, and their resolution and imaging speed have significantly improved owing to a shift from the time domain to the spectral domain method. The recently introduced swept-source OCT uses a tunable wavelength light source. This technology has higher penetration ability in deeper layers as well as in more opaque optical media [1]. As a result, OCT is now becoming an indispensable instrument in the daily practice of ophthalmology [2-9]. However, some facilities do not have permanently installed OCT devices because of their high cost and stationary nature. Additionally, taking images with conventional OCT requires patients to remain seated for a certain period of time, which makes it difficult to perform OCT examinations on patients who have difficulty maintaining their posture or transferring from one position to another. In recent years, portable OCT devices have emerged as a solution to this problem, and their usefulness in screening for age-related macular degeneration and in pediatric care has been reported [10-11]. However, most of these reports are based on qualitative evaluations, and reports on quantitative evaluations are lacking. The ACT100 ver.19 (Miis, Hsinchu, Taiwan), a lightweight and compact portable SD-OCT system consisting of a main box (234 × 190 × 108 mm, 3.35 kg) and a probe (275 × 90 × 74 mm, 0.75 kg), was launched in 2023. This OCT system is expected to be useful in clinical settings, such as medical examinations at facilities without conventional OCT equipment or during house calls, because it enables both qualitative and quantitative evaluation. However, the accuracy of this system in actual clinical situations has not been evaluated, and a comparison with existing OCT systems has not yet been conducted. In this study, we investigated the usefulness of portable OCT by examining the reproducibility of retinal thickness measurements in normal eyes using portable OCT and an existing stationary OCT (Cirrus 5000 HD-OCT plus, Carl Zeiss) and compared the results between the two models.

Images were taken until three images with a confidence coefficient automatically measured by each instrument (Cirrus: signal strength ≥ 6, ACT100: signal index ≥ 6) were obtained, and the number of images taken was compared between the two models.

Comment 2. p.3: the caption of Figure 1 should be consistent with the figure content. Is it "a b" or "A B"?

Authors’ reply: Thank you for pointing out this issue. We have added the labels “a” and “b” to the caption of Figure 1.

(Original)

Figure 1. A main box and a probe of ACT100 and the carrying case of ACT100.(Revised)

Figure 1. a. A main box and a probe of ACT100. b. The carrying case of ACT100.

Comment 3. p.6: line136, is Table 5 ? or should it be Table 6?

Authors’ reply: Thank you for pointing out this issue. Accordingly, we have changed the table number.

(Revised)

The CVs of both models showed slight variation and high stability at all sites; however, the CV of ACT100 was significantly higher than that of Cirrus (Table 5).

Comment 4. Table 6: Is the fovea thickness using ACT100 only 27.5? Compared to 258.6 in Cirrus, the difference cannot be 11.1, as Table 6 shows.

Authors’ reply: Thank you for pointing this out. The correct number was "247.5," not "27.5." We have corrected the corresponding part.

Comment 5. 4. Discussion (p.8), lines 156-157: How can the accuracy of measuring the retinal thickness of the ACT100 be defined? Since ACT100 has two times worse longitudinal resolution than Cirrus, how can authors compare the accuracy in thickness measurement, and what will the difference range be accepted in clinics? 

Authors’ reply: As you pointed out, we were interested in the accuracy of retinal thickness measurements by ACT100, which is why we conducted this study. ACT100 showed slightly greater variability than Cirrus; the reproducibility was good, and the measurements correlated well with those of Cirrus. We also asked Miis, the distributor, about the accuracy of the ACT100's auto segmentation. Based on their tests, including the comparison of approximately 1,400 B-scan images for retinal thickness calculations with manual annotations, the difference between the algorithm and manual annotation was 4.7µm +/- 4µm. Thus, the accuracy of the ACT100's auto segmentation is very high, and we believe that this data supports our results.

Comment 6. Lines 178-179: If the acquisition time for ACT100 is 9.2 seconds, and the total A-scans number (129*512, mentioned on page 2), the scan speed will be around 7k (A-scan/sec), which is not consistent with 80k shown in Table1. 

Authors’ reply: Thank you for your valuable comments. The process, taking a total of 9.2 seconds, includes auto-focus, auto-polarization adjustment, auto optical path difference adjustment, and data acquisition. For a single B-scan, scanning 770 points takes 9.63 ms. After removing the nonlinear regions at the beginning and the end, only the middle 512 points are captured. One acquisition, consisting of 129 raster scan frames, 30 horizontal line scan frames, and 30 vertical line scan frames, totals 189 B-scans and takes 1.82 seconds. Including the processing time of the MCU and the software, the time to execute an acquisition does not exceed 2.5 seconds. Based on this data, we have made the following changes:

(Original)

According to the manufacturer's announcement, the time required for the ACT100 was approximately 9.2 seconds, whereas that for the Cirrus was approximately 1.5 seconds.

(Revised)

According to the manufacturer's announcement, the time required for the ACT100 was approximately 2.5 seconds, whereas that for the Cirrus was approximately 1.5 seconds.

Comment 7. 4. Discussion (p.9), lines 201-203: the possible causes of differences in retinal thickness measurement should include differences in the "longitudinal resolution".

Authors’ reply:

Thank you for your valuable comment. Since the difference in "longitudinal resolution" might have caused the difference in retinal thickness measurements, we have made the following changes:

(Original)

Other possible causes of differences in retinal thickness measurement include differences in the scan interval, refractive index, and algorithm.

(Revised)

Other possible causes of differences in retinal thickness measurement include differences in the scan interval, refractive index, and longitudinal resolution.

Comment 8. Lines 217-219: The OMAG algorithm used by Cirrus is only related to the calculation of angiography, which will not affect the thickness measurement.

Authors’ reply: Thank you for your important comments. As you pointed out, the Optical Micro Angio Graphy algorithm and Image Gradient and dynamic programming algorithms are OCTA algorithms and were not relevant to this study. We have corrected the relevant parts as follows:

(Original)

In addition, Cirrus uses the Optical Micro Angio Graphy algorithm, whereas ACT100 uses Image Gradient and dynamic programming algorithms; this difference may also affect the measurement difference.

(Revised)

In addition, the significant difference in longitudinal resolution between ACT100 (10 µm) and Cirrus (5 µm) might also affect the measurement difference.

Comment 9. Lines 235-236: Why measurement reproducibility can be influenced by the shape of the probe?

Authors’ reply: Thank you for your question. We believe that because the device ACT100 is handheld, it may induce some rotation, reducing reproducibility. We have revised the relevant section as follows:

(Original)

One possible reason for this is the shape of the probe. According to the marketing company, this probe is being improved, and higher measurement reproducibility is expected.

(Revised)

Measurement reproducibility was slightly inferior for ACT100 compared with Cirrus. We believe that the handheld nature of ACT100 might cause some rotation, reducing reproducibility. According to the marketing company, this probe is being improved to reduce possible rotations, and higher measurement reproducibility is expected.

Comment 10. Abstract: line 17, what are "CVs"? Writing the full name before introducing an abbreviation is essential.

Authors’ reply: We apologize for not introducing the abbreviation. We have revised the text as follows:

(Original)

CVs of both models showed little variation and high stability; however, the CV of ACT100 was significantly higher than the Cirrus.

(Revised)

The coefficients of variation (CVs) of both models showed little variation and high stability; however, the CV of ACT100 was significantly higher.

Comment 11. 2.2 Image Acquisition and Evaluation: line77, "6.0X6.0 mm area" should be mm2

Authors’ reply: According to previous reports, the imaging range of OCT is often expressed as AA x BB mm. In this paper, AA mm x BB mm and AA x BB mm were not used uniformly, so we have corrected the notation as AA x BB mm. Thank you for pointing this out.

Comment 12.  4. Discussion (p.8), lines 159-162: The sentence is too long and hard to understand

Authors’ reply: Based on your suggestion, we have divided the sentence into two as follows:

(Original)

However, the number of images captured before three images with sufficient confidence coefficients were obtained was significantly higher with ACT100 in the right eye, whereas no significant difference was observed in the left eye.

(Revised)

However, the number of images captured until three images with sufficient confidence coefficients were obtained was significantly higher with ACT100 in the right eye. On the other hand, no significant difference was observed for the left eye.

Comment 13. line 176, please correct "of Cirrus".

Authors’ reply: We have made the following correction:

(Original)

The CVs of both models exhibited little variation and high stability; however, the CV of ACT100 was significantly higher than that of the other models.

(Revised)

The CVs of both models exhibited little variation and high stability; however, the CV of ACT100 was significantly higher than that of Cirrus.

Comment 14.  P.9. lines 213-215. The descriptions of the content were unclear. Is "1.379" the thickness or refractive index?

Authors’ reply:

Thank you for bringing this to our attention. “1.379" indicates a reflective index. The expression has been changed for clarity as follows:

(Original)

The refractive index is the value required to convert the optical delay measured by OCT into an equivalent thickness of 1.379 for ACT100 and is not published for Cirrus.

(Revised)

The refractive index required to convert the optical delay measured by OCT to an equivalent value is 1.379 for ACT100, which has not been reported for Cirrus.

Comment 15.  4. Discussion (p.9), line 227: Although "OCT"....., here could use "portable OCT" for comparing to existing OCT.

Authors’ reply: We have made the following correction:

(Original)

Although OCT can capture images with the same high reproducibility as existing OCT, some modifications are necessary when capturing images of patients with difficulty in fixation, narrow pupil size, elderly patients with cataracts, or other conditions that cause strong intermediate translucent zone opacities.

(Revised)

Although portable OCT can capture images with the same high reproducibility as existing OCT, some modifications are necessary when capturing images of patients with difficulty in fixation, narrow pupil size, elderly patients with cataracts, or other conditions that cause strong intermediate translucent zone opacities.

Reviewer 2 Report

Comments and Suggestions for Authors

Dear authors, I have some suggestions for improving your manuscript.

Line 29: You mentioned spectral domain OCT, you could also add a sentence about swept source technology and its higher penetration ability in deeper layers as well as in more opaque optical media. 

Line 51: Please define normal eyes. Was 20/20 vision mandatory, clear optical media, etc.? 

Line 57: Please define the Cirrus OCT, was it the 6000?

Tables 2-9: Please consider summarizing the results in 2 or 3 tables. 

Line 184: You discussed that CMT is an important criterion for treatment decisions, however, recent publications show that distinct fluid volume is a much more accurate and functionally related marker than subfield thickness. Please consider adding one or two sentences and references to your paper. Especially since AI-based image analysis could play an important role in mobile OCT devices in the future. 

Line 222: What do you mean by coronary catastrophes? I assume you mean diseases?

Line 224: For my personal interest: Is it possible to use this device as a self-monitoring OCT, or is it only used as a mobile device that requires a well-trained examiner?

Comments on the Quality of English Language

Dear authors I would suggest some small language corrections to improve readability.

For example

Abstract: Please introduce terms for CV and ICC also in the abstract.

Line 33: In addition, conventional OCT requires....

I assume you mean that using conventional OCT patients need to sit in an upright position or have to be mobile enough to be seated in front of a device... If so please rephrase the sentence accordingly that it is perfectly understandable...

Line 71: The measurement areas were defined as the fovea 1mm in diameter..., 

I had to re-read the sentence twice to understand what you mean, so please try to rephrase it to make it more clear.

Author Response

May 30th, 2024

Editor and Reviewers

Diagnostics

Thank you very much for reviewing our manuscript and providing excellent comments to help us improve the paper. We have revised the manuscript in accordance with your recommendations. Please find below point-by-point responses to the reviewers’ comments, along with corresponding changes made in the manuscript. The changes are indicated in red in the revised manuscript for your convenience.

Reviewer #2

Comment 1.  Line 29: You mentioned spectral domain OCT, you could also add a sentence about swept source technology and its higher penetration ability in deeper layers as well as in more opaque optical media.

Authors’ reply: As you pointed out, it is important to mention SS-OCT when discussing OCT. We have added this information to the Introduction section as follows:

(Original)

Since then, various companies have released various OCT devices, and their resolution and imaging speed have significantly improved owing to a shift from the time domain to the spectral domain method. OCT is now becoming an indispensable instrument in the daily practice of ophthalmology [1–8].

(Revised)

Since then, various companies have released various OCT devices, and their resolution and imaging speed have significantly improved owing to a shift from the time domain to the spectral domain method. The recently introduced swept-source OCT uses a tunable wavelength light source. This technology has higher penetration ability in deeper layers as well as in more opaque optical media [1]. As a result, OCT is now becoming an indispensable instrument in the daily practice of ophthalmology [2-9].

Comment 2.  Line 51: Please define normal eyes. Was 20/20 vision mandatory, clear optical media, etc.? 

Authors’ reply: As you indicated, we defined normal eyes as having a corrected visual acuity of 20/20 or better and not meeting the exclusion criteria. We have stated this in the revised manuscript as follows:

(Original)

Subjects were only included if they were healthy and had normal eyes.

(Revised)

Subjects were only included if they were healthy and had normal eyes, i.e., if they had a bset corrected visual acuity of 20/20 or better, and without any of the exclusion criteria.

Comment 3.  Line 57: Please define the Cirrus OCT, was it the 6000?

Authors’ reply: Thank you for your careful review. We have defined the Cirrus model more accurately as follows:

(Original)

Using two OCT systems, a portable OCT system (ACT100, MiiS, Hsinchu, Taiwan) and an existing stationary OCT system (Cirrus HD-OCT plus, Carl Zeiss, Baden-Württemberg, Germany),

(Revised)

Using two OCT systems, a portable OCT system (ACT100, MiiS, Hsinchu, Taiwan) and an existing stationary OCT system (Cirrus 5000 HD-OCT plus, Carl Zeiss, Baden-Württemberg, Germany),

Comment 4. Tables 2-9: Please consider summarizing the results in 2 or 3 tables.

Authors’ reply: Thank you for your suggestion. We have combined the tables (Table 2-5 in the revised manuscript). This change made the results much simpler and cleaner.

Comment 5. Line 184: You discussed that CMT is an important criterion for treatment decisions, however, recent publications show that distinct fluid volume is a much more accurate and functionally related marker than subfield thickness. Please consider adding one or two sentences and references to your paper. Especially since AI-based image analysis could play an important role in mobile OCT devices in the future. 

Authors’ reply: As you pointed out, distinct fluid volume has been reported as a much more accurate and functionally related marker than subfield thickness. We only evaluated retinal thickness, but we completely agree that AI-based image analysis will be required for mobile OCT devices in the future. We have included this point as a current limitation of our study.

(Revised)

In this study, only retinal thickness was evaluated. According to recent reports, distinct fluid volume measured using artificial intelligence (AI)-based image analysis is a much more accurate and functionally related marker than retinal thickness [31]. It is expected that the usefulness of ACT100 used in this study will be enhanced if distinct fluid volume can be measured using AI-based image analysis, and this is an issue for future studies.

.

Comment 6. Line 222: What do you mean by coronary catastrophes? I assume you mean diseases?

Authors’ reply: Thank you for pointing this out. Our wording was inaccurate. We have made the following changes:

(Original)

In our recent memory, there have been cases where patients were reluctant to visit hospitals due to coronary disasters,

(Revised)

In our recent memory, there have been cases where patients were reluctant to visit hospitals due to the coronavirus pandemic,

Comment 7. Line 224: For my personal interest: Is it possible to use this device as a self-monitoring OCT, or is it only used as a mobile device that requires a well-trained examiner?

Authors’ reply: Thank you for your interesting comments. Actually, the ACT100 (MiiS) that we studied in this study allows patients to acquire OCT themselves. In a few cases, we had the patients take OCT images themselves using this device, but the images were unclear. We believe that more practice is needed and that better images may be obtained when the device is stationary rather than handheld. We will consider this issue in the future.

Comment 8. Abstract: Please introduce terms for CV and ICC also in the abstract.

Authors’ reply: We apologize for not introducing the abbreviations. In the revised manuscript, one of these terms has been used more than once and the other has not. Accordingly, we have made the following changes:

(Original)

ICC showed the same high reliability as the Cirrus. CVs of both models showed little variation and high stability; however, the CV of ACT100 was significantly higher than the Cirrus.

(Revised)

The intraclass correlation coefficients showed the same high reliability as the Cirrus. The coefficients of variation (CVs) of both models showed little variation and high stability; however, the CV of ACT100 was significantly higher.

Comment 9. Line 33: In addition, conventional OCT requires....

I assume you mean that using conventional OCT patients need to sit in an upright position or have to be mobile enough to be seated in front of a device... If so please rephrase the sentence accordingly that it is perfectly understandable...

Authors’ reply:

Thank you for pointing out this issue. We have made the following changes:

(Original)

In addition, conventional OCT requires the patient to remain sitting for a certain time during imaging, making it difficult to maintain their posture or transfer to another position for imaging.

(Revised)

Additionally, taking images with conventional OCT requires patients to remain seated for a certain period of time, which makes it difficult to perform OCT examinations on patients who have difficulty maintaining their posture or transferring from one position to another.

Comment 10. Line 71: The measurement areas were defined as the fovea 1mm in diameter...,  I had to re-read the sentence twice to understand what you mean, so please try to rephrase it to make it more clear.

Authors’ reply:

Thank you for pointing out this unclarity. We have made the following changes:

(Original)

The measurement areas were defined as the fovea 1 mm in diameter of the EDTRS circle, the paracentral fovea 1–3 mm in diameter as the inner subfield, and the outer subfield 3–6 mm in diameter as the outer subfield. The inner and outer areas were divided into four quadrants: superior, temporal, nasal, and inferior in accordance with a previous report [9] .

(Revised)

We analyzed nine subfields within the ETDRS circle. The foveal central subfield was defined as the inner 1-mm-diameter circle; the pericentral subfield was located between the inner and middle 3-mm-diameter circles; and the peripheral subfield was between the middle and outer 6-mm-diameter circles. Based on a previous report, the pericentral and peripheral areas were further divided into four quadrants: superior, temporal, nasal, and inferior [12].

Round 2

Reviewer 1 Report

Comments and Suggestions for Authors

The authors have addressed the relevant issues of the paper, made changes, and provided further explanations, which is satisfactory.